# Identification of a Novel Semi-Dominant Spotted-Leaf Mutant with Enhanced Resistance to *Xanthomonas*
*oryzae* pv. *oryzae* in Rice

**DOI:** 10.3390/ijms19123766

**Published:** 2018-11-27

**Authors:** Zheng Chen, Ting Chen, Atul Prakash Sathe, Yuqing He, Xiao-bo Zhang, Jian-li Wu

**Affiliations:** 1State Key Laboratory of Rice Biology, China National Rice Research Institute, Hangzhou 310006, China; chenzheng7@126.com (Z.C.); chenting218@163.com (T.C.); atulsathe@163.com (A.P.S.); 2College of Life Science and Technology, Huazhong Agricultural University, Wuhan 430070, China; yqhe@mail.hzau.edu.cn

**Keywords:** rice, spotted-leaf, bacterial leaf blight, phytohormone, defense response

## Abstract

Many spotted-leaf mutants show enhanced disease resistance to multiple pathogen attacks; however, the mechanisms are largely unknown. Here, we reported a novel semi-dominant *spotted-leaf mutant 24* (*spl24*) obtained from an ethyl methane sulfonate (EMS)-induced IR64 mutant bank. *spl24* developed tiny brown lesions on the leaf tip and spread down gradually to the leaf base as well as the sheath at the early heading stage. The performances of major agronomic traits such as the plant height, panicle length, number of panicles/plant, and 1000-grain weight were significantly altered in *spl24* when compared to the wild-type IR64. Furthermore, *spl24* exhibited a premature senescing phenotype with degeneration of nuclear acids, significantly reduced soluble protein content, increased level of malonaldehyde (MDA), and lowered activities of reactive oxygen species (ROS) scavenging enzymes. Disease evaluation indicated that *spl24* showed enhanced resistance to multiple races of *Xanthomonas oryzae* pv. *oryzae*, the causal pathogen of bacterial leaf blight in rice, with elevated expression of pathogenesis-related genes, salicylic acid (SA) signaling pathway-associated genes revealed by real-time quantitative PCR and high-throughput RNA sequencing analysis. Genetic analysis and gene mapping indicated that the lesion mimic phenotype was controlled by a novel semi-dominant nuclear gene. The mutation, tentatively termed as *OsSPL24*, was in a 110 kb region flanked by markers Indel-33 and Indel-12 in chromosome 11. Together, our data suggest that *spl24* is a novel lesion mimic mutant with enhanced innate immunity and would facilitate the isolation and functional characterization of the target gene.

## 1. Introduction

Rice (*Oryza sativa* L.) is the major staple food for more than half of the world’s population. Safe production of rice grains is largely relied on the disease resistance levels of commercial varieties. Thus, allele mining for high level of resistance to multiple rice pathogens remains critical to rice breeders. The rice spotted-leaf or lesion mimic mutant, which spontaneously forms lesions similar to the hypersensitive response (HR) lesions in leaves, leaf sheaths and stems without obvious abiotic/biotic stresses, is one of the best sources for the elucidation of disease resistance mechanism and a potential donor for improvement of rice varieties [1].

Till now, more than 80 rice spotted-leaf mutants have been identified genetically [1,2], most of these mutants are controlled by a single recessive gene, while a few of them are governed by a dominant/semi-dominant gene or double genes. Among these mutants, *spl30* generates red-brown lesions without detectable cell death probably due to the accumulation of unknown pigments [3]. Except *spl30*, the other spotted-leaf mutants exhibit necrotic lesions with accumulation of reactive oxygen species (ROS) and cell death at/around the lesions [2,4,5]. So far, approximately 20 spotted-leaf genes have been isolated and characterized. Among them, *spl7* is the first cloned gene which encodes a heat shock transcriptional factor and involves in regulating the expression of other heat shock transcriptional factors associated with programmed cell death [6]. *Spl11* encodes an E3 ubiquitin ligase participating in the ubiquitination of unwanted proteins for 26S proteasome-mediated degradation [7]. *OsLCD1* encodes a zinc finger protein which plays the role not only in cell death but also the differentiation of calli [8]. In fact, the spotted-leaf gene-encoded products are very diversified in structures and functions and involved in nearly all aspects of life process. Broadly, these products can be classified into three categories: The first group is proteins, such as pathogenesis-related proteins involved in salicylic acid (SA) signaling pathway [9,10], CC-NB-LRR proteins responsible for innate immunity [11], eukaryotic release factor [12], eukaryotic translation elongation factor [13], RNA splicing factor [14,15], membrane-associated proteins [16,17], ion channel regulators [18] as well as clathrin-associated protein [19]. The second group is enzymes, consisting of lipid kinase [20], oxidoreductase [21], acyltransferase [22], protein kinase and mitotic-activated related kinase [23,24], cytochrome p450 monooxygenase and catalase [25], AAA-ATPase [26], and E3 ubiquitin ligase [7,27]. The third group is compounds such as fatty acids/lipids, porphyrin, and phenolic compounds [28]. The diversity and complexity of spotted-leaf gene products indicate extreme complicated mechanisms underlying the formation of leaf lesions. Therefore, identification of novel spotted-leaf mutants would be helpful and accelerate the elucidation of their molecular mechanisms for lesion formation and enhanced/decreased disease resistance.

Most of the spotted-leaf mutants show enhanced disease resistance to various major rice pathogens. Extensive studies have shown that at least 68 mutants exhibit an enhanced level of resistance to at least one type of pathogen, and 28 mutants show enhanced resistance to two types of pathogens. In fact, several mutants exhibit a broad-spectrum resistance to multiple pathogens such as *Xanthomonas oryzae* pv. *oryzae* (*Xoo*) and *Magnaporthe oryzae* [7,29] while some mutants show broad-spectrum resistance to multiple races/strains of a pathogen [5,7,30]. The enhancement of defense responses is often accompanied by the outburst of ROS and programmed cell death (PCD) [19,22,24,30]. Interestingly, ROS-triggered defense responses are often accompanied by leaf senescence which allows nutrient remobilization [19]. Furthermore, elevated expression of defense response-associated genes involved in SA, jasmonic acid (JA), abscisic acid (ABA) and ethylene (ET) signaling pathways has been observed and validated in several studies [12,31].

Here, we identified a rice spotted-leaf 24 (*spl24*) mutant from an EMS-induced IR64 (an elite indica cultivar) mutant bank. We compared the major agronomic traits, analyzed the histochemical indicators of PCD, measured the levels of chlorophylls, soluble proteins and phytohormones, evaluated the disease reaction to *Xoo*, conducted the expression analysis of defense-related genes, and determined the genetic control of the spotted-leaf phenotype. Our results demonstrated that *spl24* is a novel spotted-leaf mutant with enhanced disease resistance to multiple races of *Xoo* probably by activating the SA signaling pathway. The mutation is controlled by a semi-dominant gene (tentatively termed as *OsSPL24*) located to the long arm of the chromosome 11. The data obtained in the present study would facilitate the map-based cloning of *OsSPL24* and functional analysis of the mutation underlying the formation of lesions.

## 2. Results

### 2.1. Performance of Agronomic Traits

Under the natural summer conditions, the tiny brown lesions/spots first appeared on the leaf tips of *spl24* and spread down to the whole leaf blades approximately 13 weeks after sowing in the paddy field at the China National Rice Research Institute (CNRRI), Hangzhou, China. The leaves gradually turn into dark brown with the increasing number of lesions which also appeared on the surface of some husks in the mutant at the heading stage (Figure 1A–D). At the tillering stage, the level of chlorophyll a (Chl a) was significantly different between *spl24* and the wild-type IR64 (WT), while the levels of chlorophyll b (Chl b), carotenoid and Chl a/b ratio were similar between the two genotypes (Figure 1E). At the heading stage, the levels of Chl a, Chl b and carotenoid were all decreased significantly in *spl24* compared with WT, while the ratio of Chl a/b was similar (Figure 1F), indicating that the loss of photosynthetic pigments may be due to a reduction of the total number of living cells containing chloroplasts because of the necrotic lesions. Furthermore, the level of soluble proteins was sharply decreased in *spl24* compared with WT at the heading stage probably due to the same reason mentioned above (Figure 1G). The performance of major agronomic traits including the plant height, number of tillers/plant, panicle length, number of tillers/panicle, seed-setting rate, and 1000-grain weight were significantly altered in the mutant when compared to WT (Table 1). Taken together, our data suggested that the mutation induced the formation of brown lesions with decreasing levels photosynthetic pigments and soluble proteins especially at the heading stage and ultimately led to the altered performance of major agronomic traits in the mutant.

### 2.2. Impaired Function of Photosynthetic Capacity

Although the ratio of Chl a/b was normal, the lowered chlorophyll level in *spl24* at the heading stage might affect the photosynthetic capacity in the mutant as shown by the altered performance of agronomic traits (Table 1). To examine the photosynthetic capacity, we measured the photosynthetic parameters of leaves in WT and *spl24* at the heading stage. Our results showed that the net photosynthetic rate (*Pn*), stomatal conductance (*Gs*) and transpiration rate (*Tr*) were significantly lower in *spl24* than those of WT (Figure 2A–C), while the intercellular CO_2_ concentration (*Ci*) was similar between *spl24* and IR64 (Figure 2D). These results suggested that the lowered chlorophyll level indeed resulted in a lower photosynthetic capacity leading to the altered performance of agronomic traits in *spl24.*

### 2.3. Lesion Initiation Is Light-Dependent

Many lesion mimic mutants are light-dependent on the initiation of lesions [1]. To determine the effect of light on *spl24* lesion formation under the natural field conditions, the newly emerged leaves of *spl24* without lesions were covered with 2 cm aluminum foil for 7 days. The results showed that the shaded leaf area of *spl24* remained green without lesions similar to WT while the unshaded area had brown lesions after 7-day shading treatment (Figure 3A–D). The foil then was removed, and the light was reinstated, a few brown lesions appeared in the shaded leaf area 5 days and a large number of brown lesions appeared 15 days after removal of the foil (Figure 3E,F). No obvious changes were observed before and after shading treatment in WT (Figure 3). These results suggested that the initiation of brown lesions in the mutant was light-dependent.

### 2.4. ROS-Associated Cell Death Occurs in spl24

To determine whether cell death occurred at/around the lesions, we first carried out trypan blue staining, a traditional method for selective staining of dead cells and irreversible membrane damage [19]. The results exhibited that dark blue stains were observed at/around the lesions in *spl24* while no blue stains were detected at/around lesions in WT (Figure 4A), indicating that cell death indeed occurred in the mutant resembling the HR-induced lesions.

To further confirm the cell death in *spl24*, we measured the levels of malonaldehyde (MDA) and membrane ion leakage, two cell membrane damage indicators. The results showed that the MDA content and the value of membrane ion leakage were significantly increased in *spl24* at the heading stage compared with WT (Figure 4G,H). To examine whether DNA fragmentation, another indicator of cell death, happened in *spl24*, we carried out a terminal deoxyribonucleotidyl transferase-mediated dUTP nick-end labeling (TUNEL) assay. The results showed that a few nuclei (green) were TUNEL positive in WT, whereas a large number of nuclei were TUNEL positive in *spl24* (Figure 4J). Our results demonstrated that the mutation resulted in the damage to cell membrane and induced a large-scale DNA degeneration leading to cell death as shown by trypan blue staining.

To investigate whether the cell death was associated with the burst of ROS, we first carried out the detection of H_2_O_2_ accumulation in the leaves by 3, 3’-Diaminobenzidine (DAB) staining [32]. The results showed that the apparent red-brown precipitate was observed at and around the lesions in mutant leaves; however, no such red-brown precipitate was observed in the wild-type (Figure 4B). We further measured the H_2_O_2_ content, and the results showed that the H_2_O_2_ content in the mutant *spl24* was significantly increased compared to WT at the heading stage (Figure 4C). The results demonstrated that the appearance of brown lesions in the mutant accompanied by cell death was likely resulted from the burst of H_2_O_2_. The burst of ROS implies the disruption of ROS scavenging system in *spl24*. To test this possibility, we measured the activities of three anti-oxidative enzymes, including catalase (CAT), superoxide dismutase (SOD) and peroxidase (POD), which are usually activated to remove elevated ROS under oxidative stresses [33]. Our results showed that the CAT activity was significantly lower in *spl24* than that of WT (Figure 4D) while the activities of SOD and POD were significantly increased in *spl24* when compared to WT at the heading stage (Figure 4E,F). The results suggested that the ROS scavenging system was impaired and insufficient to remove the excessive ROS which ultimately led to the cell death in *spl24*.

It has been widely accepted that the cell death is accompanied with the altered expression of PCD-related genes [34]. To validate whether the alteration of PCD-related gene expression happened in *spl24*, we carried out the expression analysis of a set of eight metacaspase genes. Our results showed that the expression levels of *OsMC1*, *OsMC5*, *OsMC6*, and *OsMC8* were significantly increased compared with WT. In contrast, the expression levels of *OsMC2*, *OsMC3*, and *OsMC4* were apparently decreased compared with WT while the expression level of *OsMC7* was similar to that of WT (Figure 4I). Although it is uncertain which of these metacaspase plays the critical role in PCD, our results demonstrated that the mutation in *spl24* caused PCD accompanied with altered expression of metacaspase genes in the mutant.

### 2.5. Enhanced Disease Resistance to Xoo with Elevated Expression of Defense Response Genes

HR lesions prevent host plants from further invasion of virulent pathogens to nearby cells, and consequently enhance the level of disease resistance. To test whether the disease resistance level has been changed in *spl24*, we inoculated *spl24*, WT and the susceptible control IR24 using ten races of *Xanthomonas oryzae* pv. *oryzae* under the natural field conditions by a leaf-clipping method at the tillering stage. The results showed that IR24 was susceptible to all races tested while the mutant *spl24* showed significantly enhanced resistance to races HB17 (*p* ≤ 0.05), JS97-2, PXO112, Zhe173, PXO339, PXO347 and PXO349 (*p* ≤ 0.01) compared with WT, while the resistance level to GD1358, PXO71 and OS-225 was similar between *spl24* and WT (Table 2). These results indicated that the mutant displayed a broad-spectrum resistance to multiple races of bacterial blight pathogens.

Increased levels of disease resistance are often associated with up-regulated expression of defense response genes [29]. To determine the correlations between the enhanced resistance level to *Xoo* and the expression level of defense response genes in the mutant, 14 defense response genes either involved in SA or JA signaling pathway were tested by real-time quantitative PCR analysis. The results indicated that the transcriptional levels of six genes (*OsJAR1*, *OsAOS2*, *OsWRAKY45*, *OsJAZ6*, *OsJamyb*, and *OsPAD4*) involved in JA signaling pathway and eight genes (*OsPR1a*, *OsPR1b*, *OsPR3*, *OsPR4*, *OsPAL3*, *OsPAL6*, *OsCHS1*, and *OsEDS1*) involved in SA signaling pathway were apparently increased in *spl24* compared with WT (Figure 5A,B). The results clearly demonstrated that the enhanced level of disease resistance was associated with the up-regulated expression of the defense response genes in *spl24*.

### 2.6. SPL24-Mediated Disease Resistance Is Associated With the Activation of SA Signaling Pathway

We have shown that *spl24* enhanced resistance to *Xoo* is associated with the activation of defense response genes involved in both JA and SA signaling pathways. To further explore the potential resistance mechanism, we carried out high-throughput mRNA sequencing (RNA-seq) analysis. The cDNA libraries were prepared both from the leaves of 3 BC_2_F_2_-S (*spl24*-like plant) and the BC_2_F_2_-W (IR64-like plant) individual plants. The results showed that a total of 3420 differentially expressed genes (DEGs) were identified between *spl24*-like plants and IR64-like plants. Among them, 1860 genes were up-regulated, and 1560 genes were down-regulated in the *spl24*-like plants. Five SA signaling pathway-associated genes including *OsPR1a* (*LOC_Os07g03710*), *OsPR1b* (*LOC_Os01g28450*), *OsPR3* (*LOC_Os04g41620*), *OsPR4* (*LOC_Os11g37970*) and *OsPBZ1* (*LOC_Os12g36880*) were significantly up-regulated by 3.63-, 2.80-, 9.56-, 7.06- and 2.70-fold, respectively (Appendix A). In addition, the other defense marker genes involved in SA signaling pathways were all up-regulated (Appendix A). These results were similar to those in qRT-PCR. Unlike qRT-PCR analysis, the genes involved in JA signaling pathway were not detected in RNA-seq analysis. Nevertheless, our results suggested that enhanced disease resistance in *spl24* was likely resulted from the activation of SA signaling pathway.

We then performed Gene Ontology (GO) analysis to classify the functions of the 3420 DEGs identified in *spl24*-like plants. The GO term enrichment indicated that the 3420 DEGs could be classified into 29 GO terms under three biological processes with *p* ≤ 0.01. Among them, 4 GO terms belong to Functional process, one term belongs to Biosynthetic process, and 24 terms belong to Component process (Appendix A). For the 1860 up-regulated genes, a total of 23 GO terms were assigned, among them, “cytoplasmic membrane-bounded vesicle”, “cytoplasmic vesicle”, “membrane-bounded vesicle” and “vesicle” were the four most enriched and distinguished terms with *p*-values of 1.34 × 10^-21^, 1.34 × 10^-21^, 1.95 × 10^-21^ and 2.30 × 10^-21^, respectively (Appendix A). For the 1560 down-regulated DEGs, a total of 49 GO terms were assigned, among them, “ribosomal subunit”, “cytosolic ribosome” and “cytosolic part” were the three most enriched and distinguished terms with *p*-values of 8.34 × 10^-39^, 1.53 × 10^-38^ and 1.06 × 10^-37^, respectively (Appendix A). It has been shown that the secretory transport molecules generated by endoplasmic reticulum are harmful to microorganisms including pathogens. Thus, vesicle trafficking might play a critical role in the perception of PAMP during pathogen infections. Previous studies have indicated that extracellular secretory vesicle tethers can be mediated by an extracellular foam complex subunit named Exo70B2 and these vesicle tethers exhibit the immunity function against various pathogens [35,36]. Therefore, based on the results of GO enrichment, we infer that the enhanced disease resistance of *spl24* mutant is probably related to the vesicle trafficking.

To further explore the biological pathways in which *spl24* may be involved, we performed KEGG enrichment analysis for the 3420 DEGs between the *spl24*-like and IR64-like plants. The results showed that all the DEGs were classified into 11 pathways (Appendix A). The up-regulated 1860 DEGs enriched in 19 significantly pathways (*P* ≤ 0.01), among them, “Phenylalanine metabolism”, “Biosynthesis of secondary metabolites” and “Isoquinoline alkaloid biosynthesis” were the most enriched and distinguished pathways with *p*-values of 2.09 ×10^−6^, 1.05 × 10^−5^ and 1.26 × 10^−5^, respectively (Appendix A), while the 1560 down-regulated DEGs were grouped into six predominant pathways, among them, “Ribosome”, “Ribosome biogenesis in eukaryotes” and “Phenylpropanoid biosynthesis” were the most enriched and significant pathways with *p*-values of 2.82 × 10^−41^, 0.004 and 0.026, respectively (Appendix A). These results suggest that *spl24*-mediated disease resistance is likely associated with protein translation and defense response, consistent with the results of GO enrichment.

To further understand the association of *spl24*-mediated resistance and the endogenous hormone levels, we determined the levels of various hormones in the leaves of *spl24*-like and IR64-like plants. For the sake of easy comparison, the content of various endogenous hormones in the wild-type IR64 was set to 1, and the relative content of endogenous hormone in *spl24* was shown in Figure 6. The results showed that the contents of SA, JA, and ABA in the *spl24* were significantly increased by 23.2%, 62.5% and 49.7% compared to WT, respectively, whereas the IAA level was similar between the two genotypes, indicating that the broad-spectrum resistance of *spl24* was probably resulting from the activation of phytohormone-mediated pathway. Furthermore, ABA is involved in the ROS-mediated PCD [31], the increased level of ABA might contribute to the occurrence of PCD in *spl24* as shown by the positive TUNEL assay, ROS measurement and metacaspase gene expression analysis. As the up-regulated gene expression in SA signaling pathway was detected both in qRT-PCR and RNA-seq, we concluded that the enhanced resistance of *spl24* to *Xoo* was most likely resulted from the activation of SA signaling pathway.

### 2.7. Genetic Control and Physical Mapping of OsSPL24

To determine the genetic control of the spotted-leaf phenotype, *spl24* was crossed with WT IR64. All F_1_ plants derived from *spl24*/IR64 showed brown lesions on the leaves but with reduced number of lesions, indicating that the lesion mimic trait was controlled by a dominant gene(s) with dosage effect or a semi-dominant gene(s). Furthermore, the F_2_ population including 239 individuals segregated into three categories: IR64-like type, intermediate type (similar to F_1_) and *spl24*-like type. The ratio of these three types fitted to a 1:2:1 (χ^2^ = 0.78 < χ^2^_0.05_ = 3.84), thus we concluded that the spotted-leaf phenotype was controlled by a single semi-dominant nuclear gene tentatively termed as *Oryza sativa SPOTTED-LEAF 24* (Os*SPL24*).

To map *OsSPL24* in the genome, we developed a large F_2_ population by crossing *spl24* with Moroberekan, a normal green leaf japonica cultivar. Using the bulked segregant analysis approach, two simple sequence repeat (SSR) markers (RM5961, RM26908) (Figure 7A) on chromosome 11 were identified in co-segregation with the spotted-leaf phenotype of *spl24*. To fine map the gene, a total of 1386 WT individual F_2_ plants were genotyped, and the mutation was finally delimited to a 110 kb interval flanked by the markers Indel 33 and Indel 12 (Figure 7B).

Based on the Rice Genome Annotation Project (http://rice.plantbiology.msu.edu/, accessed on 30-09-2018), 19 genes are found in the 110 kb target region, including nine putative open reading frames (ORFs) (Figure 7C and Table 3), 8 retrotransposon proteins and two transposon proteins. Identification of the target *OsSPL24* responsible for the spotted-leaf phenotype is currently underway.

## 3. Discussion

In the present study, we identified a novel spotted-leaf mutant (*spl24*) which was genetically controlled by a new single semi-dominant nuclear gene located to the long arm of chromosome 11. The heterozygous F_1_ plants exhibited a fewer brown lesions than that of WT, indicating the mutation had a dosage effect on the severity of the lesion symptom. Since both the WT and mutant alleles are functional and have the effect on the formation of lesion, thus *spl24* belongs to a gain-of-function mutant. The spotted-leaf mutations usually impose significant variations on multiple agronomic traits such as the number of tillers [2] and plant height [37]. The mutant *spl24* showed significantly decreased level of photosynthetic pigments and soluble proteins as well as the plant height, seed-setting rate, and 1000-grain weight. Therefore, the *SPL24* mutation might directly affect the lesion formation and indirectly affect the performance of agronomic traits.

Besides genetic mutations, the initiation of lesions is also affected by multiple environmental factors including light, temperature, humidity, and mechanical wounding [38]. It has been shown that *Arabidopsis thaliana* lesion mimic mutant *slh1* displays leaf lesions under the condition of low temperature and humidity [39], while the rice mutant *M1009* does not show lesions when the temperature is higher than 25 °C; however, the lesions appears when the temperature is lower than 20 °C [40], in contrast, the rice *spl7* lesions are induced by high temperature (> 35 °C) and ultra-violet solar irradiation [6]. The rice *HM47* lesions are initiated under natural light [2]. Similar to *HM47*, the initiation of lesions in *spl24* is also light-dependent. These conditional mutants indicate that some specific pathways such as chlorophyll metabolism and defense signaling pathways are associated with the formation of lesions. For example, the inactivation of pheophorbide an oxygenase, a key enzyme in chlorophyll degeneration pathway, is responsible for the lesion initiation in the sorghum lesion mimic mutant *ded1* [41]. Determination of the specific pathway responsible for the lesion formation of *spl24* has yet to be characterized.

The most prominent feature of the spotted-leaf mutants is the formation of hypersensitive response-like lesions, a form of PCD. It has been shown that cell death is detected in/around the lesions of *spl11*, *HM47* and *spl33* in rice using trypan blue staining [2,7,13]. In addition, the cell death is likely caused by the burst of ROS in the mutants mentioned above as well as the rice *HM143* and sorghum *ded1* [4,41]. In the present study, *spl24* shows HR-like PCD with dosage effect. The *spl24* cell death is associated with the degradation of nuclear acid indicated by TUNEL staining, the burst of H_2_O_2_ indicated by DAB staining and the altered expression of caspase-like genes. Furthermore, ROS is the product of redox reaction and the superfluous ROS is cleaned up by the reactive oxygen scavenging system including CAT, SOD and POD to prevent cells from their harmful effects [42]. In this study, the activity of CAT decreased significantly while the activities of POD and SOD increased apparently, leading to the high level of ROS accumulation and cell death in *spl24*. It is likely that the increased activities of POD and SOD are not enough for the compensation of the lowered CAT activity for excessive ROS degeneration. However, the reasons for the altered enzymatic activities remain elusive.

Many rice spotted-leaf mutants exhibit stronger levels of resistance than their WTs to bacterial and fungal pathogens [1]. The enhanced disease resistance is associated with the upregulation of various defense response genes [2,12]. It has been shown that endogenous SA level is increased apparently, preceding the induction of pathogenesis-related genes and the onset of disease resistance [43,44,45]. In this study, the endogenous level of SA in *spl24* increased by 23.2% and the resistance of *spl24* to 8 races of *Xanthomonas oryzae* pv. *oryzae* was significantly enhanced with the upregulation of multiple defense response genes, especially the genes involved in SA pathways were detected both by qRT-PCR and RNA-seq. Taken together, this led us to concluded that OsSPL24-mediated resistance was associated with the activation of SA signaling pathway. Unexpectedly, the level of JA in *spl24* was significantly increased by 62.5%; however, the upregulation of JA signaling pathway genes detected in qRT-PCR was not detected in RNA-seq analysis due to unknown reasons. Though the interaction between SA and JA is complicated while their antagonistic relationship is well accepted [46,47]. It has been shown that accumulation of both SA and JA is associated with enhanced resistance against *Xoo* in rice [48,49]. It is likely that multiple pathways that interact with each other are involved in host resistance to *Xoo*. Nevertheless, whether JA signaling is involved in OsSPL24-mediated resistance has yet to be determined. ABA is well known for its role in triggering plant senescence [50]. Interestingly, ABA treatment inhibits both upstream and downstream signaling in the SA-mediated defense pathway, independently of the jasmonic acid/ethylene-mediated signaling pathway [51]. It seems the increased level of endogenous ABA in *spl24* did not inhibit the SA signaling as the upregulation of multiple SA-related genes was detected. Furthermore, based on the DEGs analysis, OsSPL24-mediated disease resistance was probably indirectly associated with multiple other pathways/components involved in vesicle trafficking, cell wall components, transcription factors, heat shock proteins and MAPK kinases. To elucidate the molecular mechanism of OsSPL24-mediated bacterial resistance, we are currently carrying out the map-based cloning to isolate the target gene.

## 4. Materials and Methods

### 4.1. Plant Materials and Growth Conditions

The indica rice *spotted-leaf 24* (*spl24*) mutant was isolated from an ethyl methane sulfonate (EMS)-induced IR64 mutant bank [52]. The mutant, wild-type (WT) and F_2_ populations were grown in the paddy field in the summer of 2017 at the CNRRI in Hangzhou, Zhejiang, China. The F_2_ population derived from the cross *spl24*/Moroberekan was used for genetic analysis and gene mapping. The backcrossed population derived from *spl24*/IR64 was used for high-throughput RNA sequencing analysis. The agronomic traits including plant height, number of tillers/plant, panicle length, number of filled grains/panicle, seed-setting rate and 1000-grain weight were recorded at the mature stage on three individual plants, and the means from three replicates were used for analysis.

### 4.2. Senescence-related Parameter Measurement

The upper leaves of the mutant *spl24* and WT IR64 at the tillering and the heading stages were used to measure the contents of chlorophyll a (Chl a), chlorophyll b (Chl b) and carotenoid as described previously [53]. The activities of ROS scavenging enzymes, including peroxidase (POD), superoxide dismutase (SOD), and catalase (CAT), as well as the contents of malonaldehyde (MDA), hydrogen peroxide (H_2_O_2_) and soluble proteins (SP), were determined at the heading stage following the manufacturer’s instructions (Nanjing Jiancheng Bioengineering Institute, Nanjing, China). Membrane ion leakage was determined as described previously [54]. The means from three measurements were used for analysis.

### 4.3. Shading Experiment

When the lesions appeared on the tip of the leaves in *spl24* at the early heading stage, the mutant leaves with and without lesions were shaded respectively with a piece of 2–3 cm aluminum foil for 3 days under natural light conditions. The foil was then removed, and light was reinstated for 4 days to investigate the influence of natural light on the initiation of lesions. Lesion development was documented by a scanner (HP scanner jet 4010, HP, Shanghai, China).

### 4.4. Histochemical Analysis

Leaves from WT and leaves with lesions from spl24 at the early heading stage were used to verify the cell death and H_2_O_2_ accumulation by trypan blue staining and 3,3′-diaminobenzidine (DAB) staining following the methods described previously [55,56].

### 4.5. TUNEL Experiment

For the TUNEL assay, leaves with and without lesions from the mutant *spl24* and the WT IR64 at the early heading stage were fixed by FAA solution (5% propionic acid, 40% formaldehyde and 50% ethanol) for 24 h at 4 °C. The 9µm parafilm sections of treated leaves were used for TUNEL analysis. The DeadEnd Fluorometric TUNEL system was used for nuclear DNA fragmentation assay by using a Fluorescein in Situ Cell Death Detection Kit following the manufacturer’s instructions (Roche, Basel, Switzerland). Samples were photographed using a confocal laser scanning microscope (Ceise, Jena, Germany).

### 4.6. RNA Extraction and qRT-PCR

Total RNA was isolated from the leaves at the early heading stage using NucleoZOL reagent (Macherey-Nagel, Düren, Germany), according to the manufacturer’s instructions. RNA samples were treated with DNase (Promega, Madison, USA), and then 1µg of RNA was subsequently used to synthesize first-strand cDNA for reverse transcription-PCR. The quantitative real-time PCR (qRT-PCR) was performed in a total volume of 20 μL reaction buffer containing 6 μL cDNA template (120 ng), 10μL SYBR Premix E×Taq (Takara), 4μL primer (10 μmol/μL) on a Thermal Cycler Dice Real Time System II (Takara, Kusatsu, Japan) at 95 °C pre-denaturation for 10 min and followed at 95 °C 10 s, 60 °C 30 s, for 40 cycles, with a final dissociation at 95 °C 15 s, 60 °C 30 s, 95 °C 15 s. The rice *ACTIN* was used as the internal control to normalize the expression levels. The data were analyzed by the 2^-ΔΔCt^ method and means from three replicates were used for analysis. The relevant primer sequences for defense response genes and *OsMCs* are listed in Appendix A.

### 4.7. Disease Evaluation and Phytohormone Level Determination

Rice plants of *spl24*, WT and susceptible control IR24 at the tillering stage were inoculated with 10 races (HB17, GD1358, PXO71, JS97-2, PXO112, Zhe173, OS-225, PXO339, PXO347 and PXO349) of *Xanthomonas oryzae* pv. *oryzae* (*Xoo*) by the leaf-clipping method [57]. The bacterial cultures were separately suspended in distilled water and adjusted to OD600 = 1.0 for inoculation. Five fully expanded-leaves per genotype were inoculated for each race. Lesion length was scored 20 days after inoculation. Means from five leaves were used for analysis. The levels of hormones including SA, JA, ABA, and IAA in *spl24* and WT were determined following the method described previously [12].

### 4.8. Genetic Analysis and Gene Mapping

To determine the genetic control, *spl24* was used as the female parent and crossed with the male parent IR64. The F_1_ plants and F_2_ individuals were grown in the paddy field at the CNRRI for phenotyping. To map the mutation, an F_2_ population was developed from the cross between *spl24* and the normal green leaf cultivar Moroberekan. The bulked segregant analysis approach was adopted to rapidly locate the mutation to a chromosome and then fine mapping was carried out by genotyping the WT F_2_ individuals. The DNA of the parents and F_2_ individual plants were extracted following the mini-preparation method [58]. SSR markers were obtained from the website (http://www.gramene.org/) while insertion/deletion (InDel) markers were designed using the Primer 5.0 (PREMIER Biosoft, California, USA) and DNAStar 5.0 software (DNASTAR, Madison, USA) after comparison of the sequences between the japonica cultivar Nipponbare and the indica cultivar 9311 in the public database on the website (http://gramene.org/genome_ browser/index.html, accessed on 06-11-2013). PCR reaction and detection were carried out as described previously [2]. The primers were synthesized by Sangon Biotech Co. Ltd (Shanghai, China). The relevant SSR and InDel primer sequences for gene mapping are listed in Appendix A.

### 4.9. Transcriptome Analysis

To eliminate other possible mutations presented in the mutant, *spl24* was backcrossed to the wild-type IR64. The F_2_ plants were selfed to generate BC_2_F_2_. Three individual BC_2_F_2_-S (*spl24*-like type) and three BC_2_F_2_-W (IR64-like type) plant at heading stage were chosen for isolation of total RNA. A total of 6 RNA samples were used for RNA-seq analyses. Library construction and an approximate 6G bp deep sequencing were carried out using the Illumina HiSeq 3000 platform (Illumina, San Diego, CA, USA) following the manufacturer’s instructions by Vazyme biotech co, Ltd. (Nanjing, China). The aligned read files were processed by Cufflinks, which uses the normalized RNA-seq fragment counts to measure the relative abundances of the transcripts. The unit of measurement is Fragment Per Kilobase of exon per Million fragments mapped (FPKM). DEGs between the *spl24* and IR64, based on three biological replicates of each, were identified using the Empirical Analysis of Digital Gene Expression data package in Cuffdiff (ver.2.1.1). An absolute fold change ≥ 2 and an FDR significance score ≤ 0.05 were used as thresholds to identify significant differences in gene expression. Each DEG was annotated, initially according to the top *Oryza sativa* and then by GO enrichment analysis, which was performed using GO Slim (http://www.geneontology.org) (accessed on 14-10-2017), o assign it to the one of the three principal GO categories: molecular function, cellular component, and biological process.

## Figures and Tables

**Figure 1 ijms-19-03766-f001:**
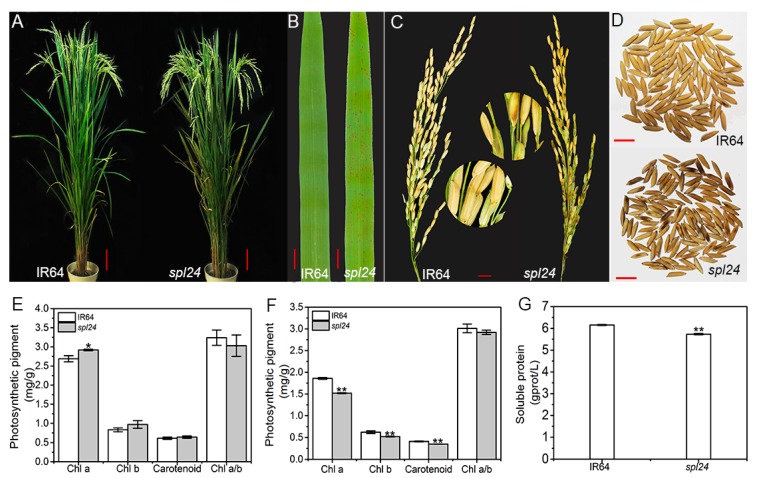
Phenotypic characteristics of *spl24* and the wild-type IR64. (**A**) Mature plant, bar = 10 cm; (**B**) Leaf of IR64 and *spl24*; bar = 1 cm; (**C**) panicle of IR64 and *spl24*, bar = 1 cm; (**D**) grains of IR64 and *spl24*; bar = 1 cm; (**E**) photosynthetic pigment contents of IR64 and spl24 at the tillering stage; (**F**) photosynthetic pigment contents of IR64 and *spl24* at the heading stage; (**G**) soluble protein content of IR64 and *spl24* at the heading stage. Values are means ± SD (*n* = 3); ** indicates significance at *p* ≤ 0.01 and * indicates significance at *p* ≤ 0.05 by Student’s *t* test.

**Figure 2 ijms-19-03766-f002:**
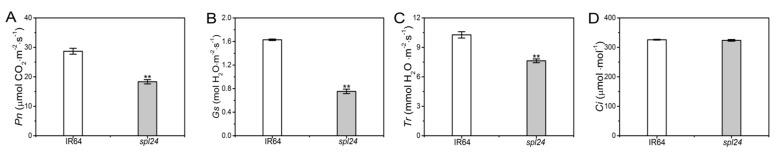
Photosynthetic parameters of leaves at the heading stage. **(A)** Net photosynthetic rate (*Pn*); **(B)** stomatal conductance (*Gs*); **(C)** transpiration rate (*Tr*); **(D)** intercellular CO_2_ concentration (*Ci*). Values are means ± SD (*n* = 3); ** indicates significance at *p* ≤ 0.01 by Student’s *t* test.

**Figure 3 ijms-19-03766-f003:**
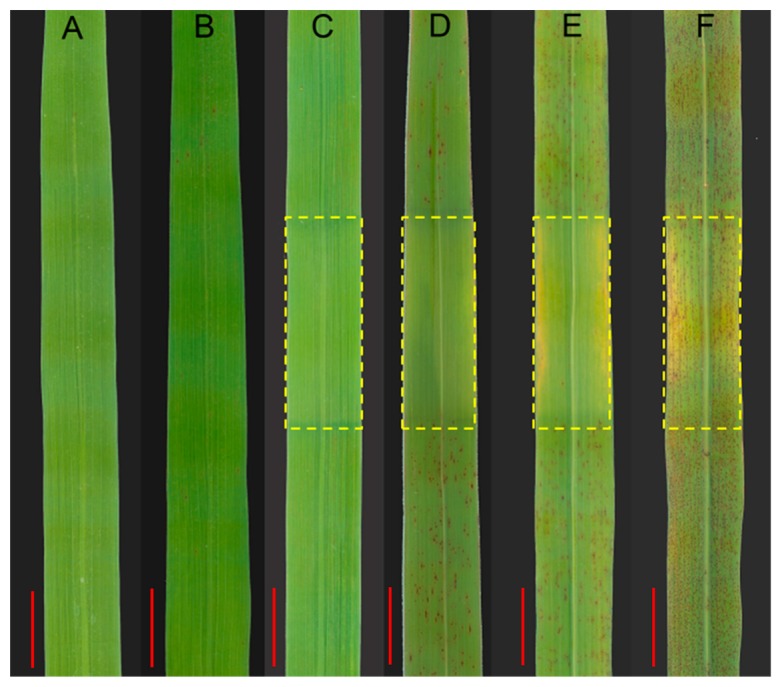
Light effect on lesion formation under the natural conditions. (**A**) IR64; (**B**) *spl24* before shading; (**C**) IR64 shaded for 7 days; (**D**) *spl24* shaded for 7 days; (**E**) *spl24* reinstated for 5 days; (**F**) *spl24* reinstated for 15 days. Shaded areas are boxed. bar = 1 cm.

**Figure 4 ijms-19-03766-f004:**
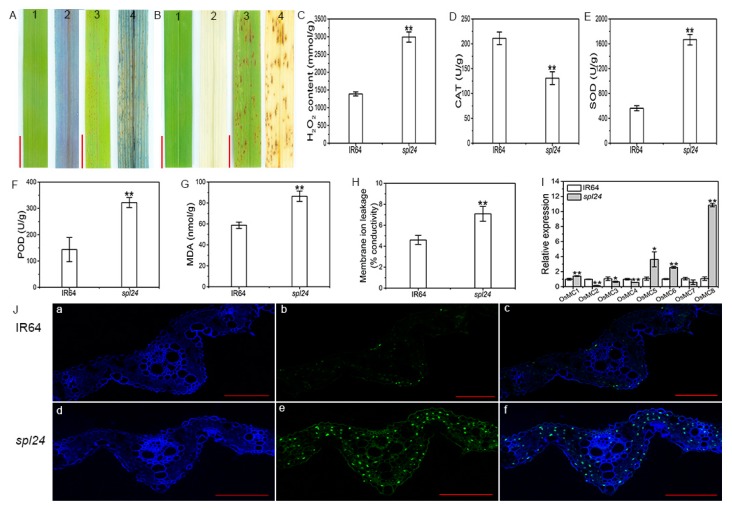
Analysis of cell death and ROS-associated parameters in IR64 and *spl24*. (**A**) Typan blue staining. 1, IR64; 2, IR64 after staining; 3, *spl24*; 4, *spl24* after staining; bar = 1 cm; (**B**) DAB staining. 1, IR64; 2, IR64 after staining; 3, *spl24*; 4, *spl24* after staining; bar = 1 cm; (C) H_2_O_2_ content at the heading stage; (**D**–**F**) the enzymatic activities of CAT, SOD, POD in the flag leaves. CAT, catalase; SOD, superoxide dismutase; POD, peroxidase; (**G**) malonaldehyde (MDA) contents of the flag leaves; (**H**) membrane ion leakage rates at the heading stage; (**I**) expression analysis of *OsMCs* at the heading stage; (**J**) TUNEL assay at the heading stage. Blue signal represents 4′,6-diamino-phenylindole (DAPI) staining; green color represents positive result. (a) and (d) are DAPI staining; (b) and (e) are TUNEL signal; (c) and f are merged images of (a/b) and (c/d) respectively, bar = 100µm. Values are means ± SD (*n* = 3); ** indicates significance at *p* ≤ 0.01 and * indicates significance at *p* ≤ 0.05 by Student’s *t* test.

**Figure 5 ijms-19-03766-f005:**
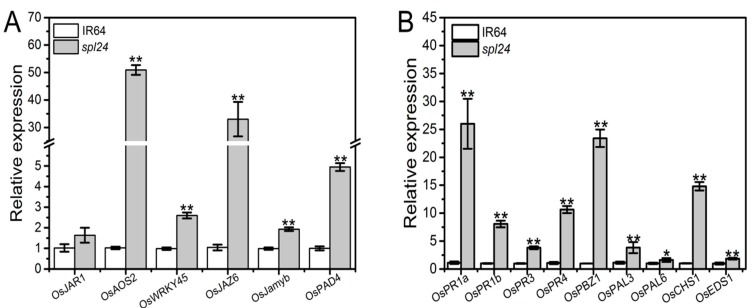
Quantitative PCR analysis of defense response genes involved in JA and SA signaling pathways. (**A**) Expression analysis of genes in JA pathway at the heading stage; (**B**) expression analysis of genes in SA pathway. The expression level of each gene in the wild-type was normalized to 1. Values indicate means ± SD (*n* = 3). * denotes *P* ≤ 0.05, and ** denote *p* ≤ 0.01 by Student’s *t* test.

**Figure 6 ijms-19-03766-f006:**
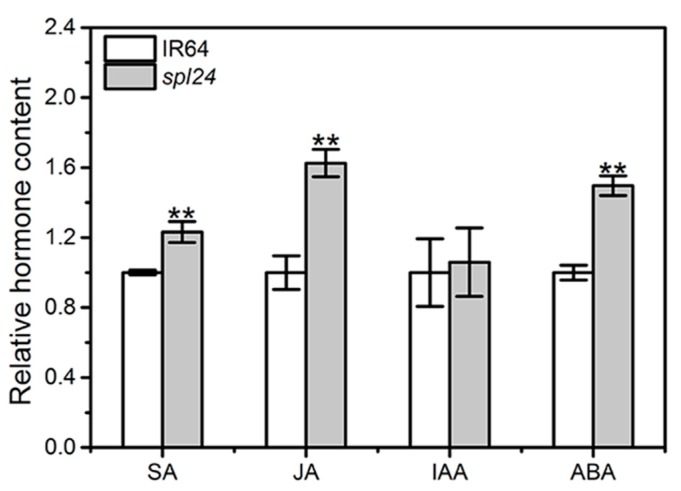
Plant hormone contents in *spl24* and IR64. The level of plant hormone in IR64 was normalized to 1. Values indicate means ± SD (*n* = 3). ** denote *p* ≤ 0.01 by Student’s *t* test.

**Figure 7 ijms-19-03766-f007:**
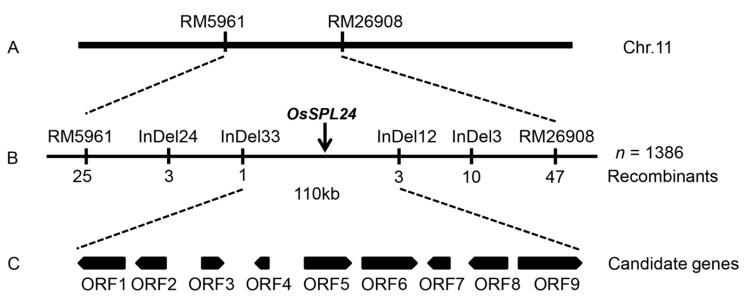
Location of the *OsSPL24* gene on the long arm of chromosome 11.

**Table 1 ijms-19-03766-t001:** Performance of agronomic traits in IR64 and *spl24.*

Material	Plant Height (cm)	No. Tiller/Plant	Panicle Length (cm)	No. Filled Grain/Panicle	Seed-Setting (%)	1000-Grain Weight (g)
IR64	117.0 ± 0.5	19.3 ± 3.2	26.4 ± 0.2	58.0 ± 3.6	50.4 ± 1.1	24.3 ± 0.2
*spl24*	109.8 ± 1.3 **	28.0 ± 1.0 *	25.6 ± 0.6 *	69.3 ± 2.5 *	55.6 ± 0.6 **	22.9 ± 0.3 **

Values are means ± SD (*n* = 3); ** indicates significance at *p* ≤ 0.01 and * indicates significance at *p* ≤ 0.05 by Student’s *t* test.

**Table 2 ijms-19-03766-t002:** Evaluation of disease resistance to *Xanthomonas oryzae* pv. *oryzae*.

	Lesion length (cm)		
race	IR24	IR64	*spl24*
HB17	25.39 ± 3.85	5.71 ± 1.06	3.70 ± 1.39 *
GD1358	12.87 ± 0.51	5.60 ± 1.28	4.40 ± 1.68
PXO71	23.10 ± 4.71	3.05 ± 0.33	2.43 ± 0.93
JS97-2	16.48 ± 2.29	12.40 ± 1.41	7.00 ± 1.96 **
PXO112	23.93 ± 2.86	4.82 ± 0.88	3.40 ± 0.89 **
Zhe173	19.98 ± 3.07	7.01 ± 1.02	4.80 ± 1.43 **
OS-225	14.32 ± 2.92	2.41 ± 0.67	2.40 ± 0.57
PXO339	22.14 ± 2.32	16.63 ± 2.01	9.20 ± 1.95 **
PXO347	21.40 ± 3.67	18.15 ± 2.11	10.10 ± 1.66 **
PXO349	17.68 ± 2.71	13.17 ± 1.80	9.40 ± 2.20 **

Values are means ± SD (*n* = 3); ** indicates significance at *p* ≤ 0.01 and * indicates significance at *p* ≤ 0.05 by Student’s *t* test.

**Table 3 ijms-19-03766-t003:** The annotation of candidate genes.

ORF	Gene ID	Annotation
ORF1	LOC_Os11g34460	OsFBO10-F-box and other domain containing protein, expressed
ORF2	LOC_Os11g34470	expressed protein
ORF3	LOC_Os11g34570	lysM domain containing GPI-anchored protein precursor, putative, expressed
ORF4	LOC_Os11g34580	hypothetical protein
ORF5	LOC_Os11g34610	DUF26 kinases have homology to DUF26 containing loci, expressed
ORF6	LOC_Os11g34624	DUF26 kinases have homology to DUF26 containing loci, expressed
ORF7	LOC_Os11g34640	expressed protein
ORF8	LOC_Os11g34650	expressed protein
ORF9	LOC_Os11g34660	Protease inhibitor/seed storage/LTP family protein precursor, expressed

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
