# Peer review of "Identification of a Novel Semi-Dominant Spotted-Leaf Mutant with Enhanced Resistance to Xanthomonas oryzae pv. oryzae in Rice"

_ijms, 2018, doi:10.3390/ijms19123766_

Round 1
Reviewer 1 Report
The manuscript by the Chen et al is very important study to identify the mutant rice against Xanthomonas pathogen.
There are some points to clarify before considering this manuscript for the publication.
1. In the Materials and Methods section, the experimental procedures for the expression analysis of OsMCs and the enzyme activities are missing. Overall the methodology section needs more better explanation.
2. There is no explanation of how many biological replicates were used.
Rest the manuscript needs the thorough checking by the native English speaker.
Author Response
1. In the Materials and Methods section, the experimental procedures for the expression analysis of OsMCs and the enzyme activities are missing. Overall the methodology section needs more better explanation.
Response: Thank you very much for pointing out the mistake. We have included the procedure for OsMCs as well as defense gene expression qRT-PCR analysis. More detailed explanation for various methods including the determination of enzyme activity, MDA, membrane ion leakage, hydrogen peroxide and phytohormone level are provided in the newly revised manuscript.
2. There is no explanation of how many biological replicates were used.
Response: Thanks again. We have carefully checked through the whole text including the methodology section, table captions and figure legends in the revised manuscript.
Rest the manuscript needs the thorough checking by the native English speaker.
Response: Dr. Lei Wang is one of our colleagues and is fluent in English. He is also a long-term English editor of Rice Science. Upon our request, He kindly helps us in language editing of our revised manuscript.
Reviewer 2 Report
The manuscript by Chen et al. describes the identification of a novel semi-dominant mutant in rice with enhanced resistance to a pathogen which is accompanied by reduced photosynthetic capacity and slight growth trade-offs.
The manuscript is very well written and the results are well and clearly presented. I have only minor suggestions to improve presentation of some results.
Lines 45 onwards. I suggest to start with a description of how the mutant was obtained/isolated? Was it identified from a larger screen or was it specifically selected?
Page 5, second paragraph: The order of the panels in Figure 4 does not match their appearance in the text. Panels 4G and H are mentioned before panels B, C, etc.
Page 6, Figure 4: panels A and B are too small and should be increased in size.
I miss a discussion of the candidate genes in the discussion section? What is known about them or related family members?
Author Response
The manuscript is very well written and the results are well and clearly presented. I have only minor suggestions to improve presentation of some results.
Response: We are grateful to your overall comments to our manuscript.
Lines 45 onwards. I suggest to start with a description of how the mutant was obtained/isolated? Was it identified from a larger screen or was it specifically selected?
Response: In the revised version, we simply state in line 34 that this mutant was identified from an EMS-induced IR64 mutant bank. The mutant spl24 was identified because of the appearance of lesions on its leaves.
Page 5, second paragraph: The order of the panels in Figure 4 does not match their appearance in the text. Panels 4G and H are mentioned before panels B, C, etc.
Response: Thank you very much for the suggestion. The reason we mentioned 4G/H before 4B/C is that we want to make the whole Figure4 look more pleasant in the current arrangement. We have tried several other arrangements, and only the current one is the best although it does not follow the Arabic order.
Page 6, Figure 4: panels A and B are too small and should be increased in size.
Response: Thanks for the suggestion. We make both panels A and B slightly bigger and replace the old Figure 4 in the revised version.
I miss a discussion of the candidate genes in the discussion section? What is known about them or related family members?
Response: We are currently sequencing all the ORFs in the target region. After that, the complementation experiment will be carried out to validate a candidate ORF. Thus, at the moment, we are not sure the nature of the suspected candidate gene. We aim to publish another paper dealing with the function of OsSPL24 in the future.
Reviewer 3 Report
The manuscript is very well written, clear and easy to read. The authors make an exhaustive comparison between the mutant found and the controls. From my point of view, it is aceptable.
Author Response
The manuscript is very well written, clear and easy to read. The authors make an exhaustive comparison between the mutant found and the controls. From my point of view, it is aceptable.
Response:We are really grateful to your comments. In the revised version, we present in more details for the explanation of relevant methods and results that mentioned by other reviewers.